# Associations between 24-h Movement Behavior and Internet Addiction in Adolescents: A Cross-Sectional Study

**DOI:** 10.3390/ijerph192416873

**Published:** 2022-12-15

**Authors:** Caizhen Ma, Jin Yan, Hejue Hu, Chongyan Shi, Feng Li, Xinyue Zeng

**Affiliations:** 1School of Physical Education, Shanxi Normal University, Linfen 041081, China; 2Centre for Active Living and Learning, University of Newcastle, Callaghan, Newcastle, NSW 2308, Australia; 3College of Human and Social Futures, University of Newcastle, Callaghan, Newcastle, NSW 2308, Australia; 4Library of Beijing Sport University, Beijing 100084, China; 5School of Physical Education and Humanity, Nanjing Sport Institute, Nanjing 210014, China; 6China Basketball College, Beijing Sport University, Beijing 100084, China; 7Faculty of Physical Education, China West Normal University, Nanchong 637000, China

**Keywords:** internet addiction, 24-h movement behavior, adolescents, youths, cross-sectional study

## Abstract

Objective: This study aimed to explore the relationship between 24-h activity behavior and Chinese adolescents’ Internet addiction. Methods: A survey of 2045 adolescents (56.5% boys) was conducted, and adolescents’ 24-h movement behavior and Internet addiction were measured via a questionnaire. Descriptive statistics were used to analyze the basic situation of the respondents; chi-square analysis was used to compare gender differences, and logistic regression was used to analyze the relationship between 24-h exercise guide entries and Internet addiction. Results: From the number of 24-h movement behavior guides, 25.3% of the children did not meet the recommended amount of any kind of activity behavior guide, while 50.4% and 21.7% of the children reached the recommended amount of one and two activity behavior guidelines, respectively; only 3.2% of the children met the recommended amount of all three activity behavior guidelines. Adolescents who did not meet the recommended 24-h activity behavior guidelines were more likely to have Internet addiction (OR = 8.46, 95 CI = 3.06–23.36), and were more likely to have one item (OR = 4.50, 95 CI = 1.64–12.39) or two items (OR = 3.12, 95 CI = 1.11–8.74). Conclusions: Physical activity, static behavior, and sleep may all have a greater impact on adolescents’ Internet addiction, among which physical activity has a greater impact on adolescents’ Internet addiction. Different combinations of 24-h movement behavior have different effects on adolescents’ Internet addiction.

## 1. Introduction

The Internet is the most frequently used recreational and academic tool by adults and adolescents. A virtual world offers people an easy and immediate way to access information and communicates with other people all over the world [1]. With significant internet use and a significant increase in online access, internet addiction has become more frequent in society [2]. In the 47th most recent statistics on Internet development in China, 989 million people used the Internet by December 2020 [3], of these people, 14.8% were aged 10 to 18, showing that the Internet has become an inseparable part of adolescents’ lives [4]. Internet addiction (IA) can be defined as a cognitive control disorder that does not involve specific drugs [5]. Based on the clinical perspective, IA can be described in different variations, such as “Internet addiction disorder” [6], “pathological Internet use” [1], and “problematic Internet use” [7]. This clinical issue has been studied by several researchers. For example, as a result of Young’s modification of the Diagnostic and Statistical Manual of Mental Disorders, the 4th edition (DSM-IV), pathological gambling has been defined differently [8], developing diagnostic questionnaires for pathological Internet usage. A conceptual framework based on impulse-control disorders in the Statistical Manual of Mental Disorders (DSM-IV-TR) is also used by Shapira et al. to diagnose problematic Internet use [7]. 

According to previous studies, excessive internet usage can lead to internet addiction, which can decrease academic performance [9], social isolation [10], and emotional disorders [11]. Moreover, severe internet addiction can also lead to social problems, such as deteriorating parent-child relationships [12], self-harm [13,14], and increased delinquency and crime rates [15], bringing severe harm to families, schools, and society. Therefore, paying attention to the psychological and social adjustment of Internet addicts among adolescents is essential.

Past studies have shown a negative correlation between physical activity levels and internet addiction, with internet-addicted adolescents showing less physical activity [16,17]. In addition, there is a link between insomnia, apnea, nightmares, and mental health difficulties in people with IA [18]. Regarding physical health, IA is associated with a higher body mass index, resulting from prolonged sedentary behavior in adolescents [19,20]. A recent study has shown that sedentary behavior is associated with young adults’ physical activity and cell phone addiction [21]. The academic concept of “24 h movement behaviors” has recently emerged internationally. As the full spectrum of movement intensity over 24 h, sleep, sedentary behavior, and physical activity are collectively referred to as “24-h movement behaviors” [22,23,24]. The 24-h guidelines emphasize that a holistic study of 24-h movement behavior is the only way to provide a more three-dimensional picture of opportunities to improve adolescent physical and mental health [25,26,27].

There has been no comprehensive study on the association between the relative distribution of 24-h action behavior time and Internet addiction. Previous studies have mostly emphasized the independent effects of physical activity, sedentary time, and sleep on Internet addiction in adolescents. Therefore, our analysis of Chinese adolescents aimed to understand the relationship between 24-h movement behavior and Internet addiction, and to provide a basis for health promotion in this population.

## 2. Methods

### 2.1. Study Design and Participants

In the current study, the relationship between 24-h movement behavior and health fitness and psychological health of children in was studied, and primary school students in grades 4–5 in the main city of Shanxi were selected as subjects (the survey was limited to the main city due to the new crown epidemic). This study selected 4th–5th-grade elementary school students as the subjects, mainly based on the following reasons: (1) reference to the international HBSC (health behavior in school-aged children) for the selection of children’s age/school segment [28]; (2) several key indicators of this study were obtained from the questionnaire, and the reading and comprehension skills of elementary school grades 1–3 are still immature and are prone to making many errors in understanding and responding to the questionnaire; therefore, children in grades 1–3 were not included in this study. In addition, students in grade 6 of elementary school face the entrance examination for primary school, and their academic burden is heavy, making it difficult for them to cooperate with the test and carry out the intervention experiment; hence, students in grade 6 were not included in this study. The findings of this study provide more scientific advice on 24-h activity behaviors for children in this period and help children obtain more health benefits during their growth spurt.

### 2.2. Procedures

In this study, schools in different main urban areas of Shanxi were contacted and invited to participate in this survey using convenience sampling (influenced by the epidemic) and considering regional and economic level differences in urban areas of Shanxi. Eight schools (2–3 schools in each district) were contacted in the four main urban areas of Shanxi, where the survey was conducted for students in grades 4–5. In each school, classroom teachers were asked to assist with the survey, and 5–6 classes (30–40 students in each class) were randomly selected in each grade. A total of 3125 students were contacted for the survey. In the end, a total of 2045 students volunteered to participate in this study and completed valid questionnaires. During the survey, questionnaires were distributed and collected on-site by the research investigator who entered the school. At the same time, the researcher used on-site supervision to answer students’ questions during the questionnaire completion process, to ensure the quality of the questionnaire. After all, students completed the questionnaires, the research investigators collected and reviewed the questionnaires to ensure their completeness.

### 2.3. Measurements

An Overview of Demographics and Socioeconomics

A student self-reported questionnaire collected demographic information about the children and adolescents, including gender (boy or girl), grade (4, 5, … 12). Socio-economic status (SES) includes parents’ education level (junior high school and below, including middle school; high school/junior college/vocational high/technical school; university/college/higher education; master’s or doctorate; do not know), and socioeconomic status (family well off). 

### 2.4. 24-h Movement Behavior

#### Assessment of PA

In this study, physical activity (PA) was measured using a reliable and valid item derived from the Health Behavior in School-aged Children survey questionnaire (reliability coefficient = 0.82) [28]. In the past week, how many days did you participate in MVPA for at least 60 min on weekdays? (0 = none, 1 = 1 day, 2 = 2 days, 3 = 3 days, 4 = 4 days, 5 = 5 days, 6 = 6 days, and 7 = 7 days). To meet the moderate-to-vigorous physical activity (MVPA) guidelines, participants must report 7 days with a minimum of 60 min of MVPA daily, according to the Canadian 24-h Movement Guidelines [22].

### 2.5. Assessment of Sedentary Behavior

The sedentary behavior (ST) was also measured using reliable and valid items from the Health Behavior in School-aged Children questionnaire [28]. (1) How many hours did you spend watching TV or movies in your leisure time on weekdays and weekends over the past week? (reliability coefficients: 0.74 and 0.72, respectively); (2) On weekdays and weekends over the past week, how many hours did you spend playing video games in your leisure time? (reliability coefficients: 0.54 and 0.69, respectively); and (3) During the past week, how many hours did you spend using electronic screens for leisure activities during weekdays and weekends? (reliability coefficients: 0.33 and 0.50, respectively). Answers to these questions could be none, about 0.5 h, 1 h, 2 h, or 3 h. The Canadian 24-h Movement Guidelines recommend 2 h of ST per day to meet the ST guideline [22].

### 2.6. Assessment of Sleep

Sleep duration (SLP) was measured by 1 item from the China Health and Nutrition Survey, which has accepted validation (reliability coefficient = 0.83) [29]. Since naps do not exist in Shanxi, sleep time is measured using the Pittsburgh Sleep Quality Questionnaire on a single day’s sleep of children. Relevant measurement indicators have been reported by previous researchers, indicating that there is sufficient reliability and validity, making it suitable for the investigation of the sleep time of children and adolescents in China.

### 2.7. Internet Addiction

Internet addiction was assessed using 10 Internet addiction tests that were translated into Chinese [30]. The scale has a total of 10 questions, each of which is divided into two Likert-type formats (0 = no, 1 = yes). Participants were required to report their symptoms on a scale over the past year, with a total score ranging from 0 to 10, with an overall score above 4 indicating internet addiction. The Chinese version of the Internet Addiction Scale has been used many times in Chinese adolescents and has been shown to have good reliability validity with a Cronbach’s alpha of 0.82. Based on the assessment, study participants were divided into two groups of with IA or not with IA.

### 2.8. Statistical Analyses

Descriptive statistics were used to analyze the basic conditions of the subjects, such as the indicators of socio-demographic characteristics of the sample including gender, grade, whether the sample was an only child, the educational level of the parents, the attainment of the recommended amount of 24-h exercise behavior guidelines, and the basic conditions of Internet addiction. Descriptive statistics of 24-h activity, activity guideline attainment, and the basic profile of time spent in physical activity, static behavior, and sleep of the study subjects, were expressed in terms of mean ± standard deviation (M ± SD).

Differences in variables by gender and socio-economic characteristics were compared using chi-square analysis, and the relationship between 24-h motor behavior and Internet addiction was analyzed using the Pearson correlation coefficient. Logistic regression analysis was used to analyze the relationship between 24-h exercise guideline entries and Internet addiction by using BMI, gender, school band, region, and parental education of the study subjects as control variables, with the attainment entries and combinations of 24-h movement behavior used as independent variables, respectively.

## 3. Results

The basic profile of the sample included in this section is shown in Table 1, with data from 2045 children. The mean index of affluence of the sample families was 4.9 ± 1.7. There were 56.5% boys (n = 1156), 45% were in the fourth grade of elementary school (n = 921), 25% were only children (n = 511), and 94.7% were living with their parents. The education level of the sample parents was mostly concentrated in the categories of junior high school and below (including junior high school), high school/junior high school/vocational high/technical school, and university/college/higher vocational, with 86.1% of the fathers’ education level in these three categories, and 87.7% of the mothers’ education level in these three categories.

Table 2 shows that from the perspective of guidelines for individual activity behaviors, the proportion of screen time guides recommended was the highest 61.9%, followed by sleep guidelines and MVPA guidelines, accounting for 28% and 11.6%, respectively. From the number of 24-h movement behavior guides, 25.3% of children did not meet the recommended amount of any kind of activity behavior guide, 51.2% and 20.3% of children reached the one-item recommended amount of one and two activity behavior guidelines, respectively, and only 3.2% of children met the recommended amount of all three activity behavior guidelines. From the combination of 24-h activity behavior guidelines, the proportion of children who reached both sleep and screen time guides recommended amounts was 14.3%, and the proportion of children who also achieved sleep and moderate- to high-intensity physical activity guidelines was 0.9%, while screen time was reached. The proportion of children with the recommended amount of moderate- to high-intensity physical activity guidelines was 5.1%.

Table 3 shows 24-h activity behavior guidelines (quantity) and logistic regression results for Internet addiction. Taking the combination of sleep, screen time, and MVPA as a reference, the probability of Internet addiction in children and adolescents who reached the two recommended amounts was 3.12, and the results were statistically significant (95% CI: 1.11–8.74); the combination of the three variables of sleep, screen time, and MVPA all reached the statistic. Children and adolescents who reached a recommended amount were 4.50 times more likely to have an Internet addiction, and the results were statistically significant (95% CI: 1.64–12.39); based on the statistic that the combination of the three variables of sleep, screen time, and MVPA all reached the statistic, the probability of Internet addiction in children and adolescents without a standard reached the recommended amount was 8.46, and the results were statistically significant (95% CI: 3.06–23.36).

Table 4 shows the relationship between the 24-h activity behavior guide (different combinations) and Internet addiction. Taking the children and adolescents who reached the recommended amount of sleep, screen time, and MVPA as a reference, the probability of Internet addiction in children and adolescents who reached screen time and MVPA recommendation was 4.33, and the results were statistically significant (95% CI: 1.45–12.87); children and adolescents who achieved the recommended amount of sleep, screen time, and MVPA recommendation was referenced. Children and adolescents who achieved sleep time and recommended amounts of intermediate and higher physical activity were 5.62 times more likely to have an Internet addiction, and the results were statistically significant (95% CI: 1.57–20.16). The results of children and adolescents who achieved the recommended amount of sleep, screen time, and MVPA recommendations were not statistically significant. The results of children and adolescents with Internet addiction who achieved the recommended amount of sleep and screen time were not statistically significant. Taking the children and adolescents who reached the recommended amount of sleep, screen time, and MVPA as a reference, the probability of Internet addiction in children and adolescents who reached the standard of medium- and high-intensity activity was 7.29, and the results were statistically significant (95% CI: 2.37–22.40). For children and adolescents who reached the recommended amount of sleep, screen time, and MVPA, the probability of children and adolescents meeting the sleep standard of Internet addiction was 4.05, and the results were statistically significant (95% CI: 1.42–11.56) In the case of children and adolescents who reached the recommended amount of sleep, screen time, and moderate- and high-intensity physical activity, the probability of Internet addiction in children and adolescents without reaching sleep, screen time, and medium- and high-intensity activity was 8.44, and the results were statistically significant (95% CI: 3.06–23.31). The fact that adolescents who reach the only recommended amount of MVPA per day or achieve the recommended amount of both sleep and physical activity demonstrate a relatively high probability of Internet addiction (7.29 and 5.62 correspondingly) in comparison with those who met none of the three guidelines for daily individual activity behaviors (8.44). As well, those who meet the recommended amount of sleep and screen time have the lowest probability of Internet addiction (2.35). The Odds ratio obtained allows us to propose the hierarchy of recommended priorities for health promotion campaigns focusing on protective and harmful compositions of movement behaviors.

## 4. Discussion

Achievement of the recommended 24-h movement behavior guidelines in this study was similar to that of previous studies, and very few children were able to meet all three activity behavior guidelines at the same time. Hui used questionnaires to survey 12,590 students in eight major Asian cities (Bangkok, Hong Kong SAR, Kuala Lumpur, Seoul, Shanghai, Singapore, and Taipei) about the prevalence of adolescents reaching the 24-h activity behavior guide [31]. The results showed that about half of Asian adolescents did not meet any of the three indicators in the 24-h activity behavior guidelines, and less than 1% of adolescents, except Shanghai (3%), met the recommended levels of all three guidelines. Similar results have been reported among adolescents in Asian countries such as Hong Kong [32] and South Korea [33], as well as in other countries and regions such as Canada. Tapia-Serrano et al. found, in 23 countries and regions, that only 7.12% reached the recommended amount of the three activity behavior guidelines in the 24-h movement behavior guide, and 19.21% failed to meet the recommended amount of any of the three activity behavior guidelines [25]. Among different regions, 17.20% of the three activity behavior guidelines were achieved in Africa, 3.80% in Asia, 9.62% in Europe, 7.88% in North America, and only 2.93% in South America, with South America having the lowest proportion of 24-h movement behavior guidelines met. These findings suggest that excessive screen time, physical inactivity, and lack of sleep are common among children and adolescents globally.

This study showed that only 3.2% of children could meet all three activity behavior guidelines at the same time, which is similar to the other Asia-related research results mentioned above and is significantly lower than other regions such as Europe, Africa, and North America, indicating that Asian children generally have worse 24-h movement behavior completion than other regions such as Europe. Therefore, effective policies should be implemented as soon as possible to help children and adolescents in our country develop a healthy lifestyle, such as providing more facilities for physical activity in urban parks, playgrounds, and other community environments; increasing family publicity, such as limiting children’s screen time and sedentary behavior; encouraging children to actively participate in various types of physical activities; ensuring sufficient physical activity time; and developing regular sleep habits. The results of this study showed that the proportion of moderate- to high-intensity physical activity guidelines met was the lowest among the individual activity behaviors. Studies have found that only about one in ten students meet current guidelines for moderate- and vigorous-intensity physical activity, with boys and young children reporting higher levels of physical activity than girls and older children [34]. Although 90 percent of children reported participating in sports in the past year, the percentage of children who met physical activity guidelines was low. This result confirms the latest conclusion of the 2018 Chinese Physical Activity Report Card that existing physical activity participation may not be sufficient to ensure that children meet the recommended levels of physical activity guidelines [35]. Due to their special identity, adolescents spend most of their lives on campus, and most of the responsibility for promoting students’ physical activity is attributed to school sports or physical education. However, it should be noted that physical education with professional guidance is not possible every day or all year round; children still need to take cultural classes most of the time in school, and sports activities after school are limited. Attributing pressure to promote physical activity in schools through sports can cause families and communities to lose sight of their shared responsibilities. Therefore, measures to promote physical activity for children should not be limited to physical education classes; other initiatives are necessary to encourage students to engage in physical activity in the school environment, including appropriate strengthening of the volume and intensity of physical activity in physical education classes, strong support for physical activity between, before, and after school, and encouragement of multi-stakeholder participation in family sports and community sports.

In addition, the achievement rates for the physical activity and sleep duration guidelines were the lowest in this study, suggesting that future interventions should be targeted to children’s physical activity and sleep behavior to improve the overall achievement of the 24-h activity guidelines. In terms of single-activity behavior, screen time guidelines have the highest rate of reaching the recommended amount. Screen time achievement data varied across some studies, possibly due to differences in the devices that were included in screen time between studies. Some studies included screen time only for TVs, video games, and computers, while some of the newer studies also included media devices such as tablets and smartphones [36]. Therefore, future research should consider including all forms of screen time, as much as possible, to guarantee accurate measurement of screen time.

The analysis showed that whether two recommendations are reached, or one, or even whether the three variables’ recommendations are not reached, it is still possible for children and adolescents to have an Internet addiction. This is related to the family background and school background of adolescents. Beginning from the family level, many parents in modern society are busy with work and have no way to provide their children with sufficient discipline. Even if the elderly are at home to take care of the children, in the case of the guardian being the elderly, the discipline of adolescents may not be in place, which will facilitate adolescents becoming addicted to the Internet [37]. Another point worth noting is that when adolescents are in the advanced stages of school, the focus of teachers at this time is on the learning of adolescents, thus ignoring the sleep and physical activity requirements of children and adolescents. Under these various pressures, children and adolescents have the possibility of becoming Internet-addicted. At the same time, prolonged time spent engaging in static behaviors may cause feelings of isolation, and can consequently have adverse effects on mental health. It has been shown that adolescents with higher static behaviors (e.g., time spent watching TV and playing games) may lead to social isolation, and therefore, mental health problems [21]. In this study, we also found that sleep was associated with Internet addiction, so we can infer that PA, screen-based SB, and sleep may influence Internet addiction through negative emotions. Furthermore, screen-based SB may displace time that could otherwise be spent on PA and sleep [38], and lead to a loss of benefits of PA and sleep.

Internet addiction has a very important impact on the physical and mental health of children and adolescents. The analysis showed that in the 24-h activity behavior of children and adolescents, children and adolescents may not have the probability of becoming Internet-addicted only when sleep and screen time are up to standard. There is no direct relationship between moderate-to-high levels of physical activity, sleep time, and whether children and adolescents will have Internet addiction to the recommended standards [39]. The reason for this may be that when children and adolescents reach enough sleep time and have appropriate screen time, they achieve a certain sense of satisfaction in their hearts, which will reduce the probability of Internet addiction. On the contrary, when there is enough sleep and enough middle- and high-level physical activity, too much screen time will exacerbate the dependence of children and adolescents on electronic devices [36], and too little screen time will also make children and adolescents feel unsatisfied, thereby increasing the probability of Internet addiction [40]. PA can reduce the symptoms of depression and anxiety in adolescents to some extent, and improve their self-esteem and mental health [41]. Therefore, the positive impact of PA on Internet addiction may also be mediated by psychological function. Its positive effect on IA may be mediated by general psychological functions.

The combination of PA and screen-based SB may be related to IA through biological, psychological, and behavioral mechanisms. PA may have a positive effect on the structure and function of the brain; therefore, adolescents who participate in physical activity for long periods are less likely to be prone to Internet addiction. One study showed that after eight months of after-school training for sports, scans using functional magnetic resonance imaging revealed increased activation in several areas that support active performance [42]. It has also been shown that long-term participation in PA promotes brain white matter integrity [43], and also improves adolescent performance in inhibition tasks and flexibility tasks [44]. However, static screen-based behaviors may impair brain structure and function. Some studies have shown that screen time is associated with reduced connectivity between brain areas that are related to language and cognitive control [45]. It is possible for many factors to impact the adolescent brain, which undergoes profound changes. Earlier studies on behaviors related to Internet addiction have shown that there is reduced cortical thickness in the orbitofrontal cortex and gray matter in the anterior and posterior cingulate cortex in adolescents addicted to the Internet, dorsolateral prefrontal cortex, orbitofrontal cortex, and insula [46,47,48]. Additionally, reduced functional connectivity in circuits connecting cortical and subcortical areas—the dorsal striatum—plays a significant role. [49]. Thus, it can be suggested that physical activity, static behavior, and sleep have the potential to alter the structure and function of the brain which, in turn, can have an impact on adolescents’ Internet-addictive behavior [50]. In the future, adolescents should be actively encouraged to participate in physical activity, reduce static behaviors, and sleep more, to prevent and reduce the likelihood of Internet addiction.

## 5. Conclusions

In summary, we found that the prevalence of Internet addiction among adolescents in China is relatively high. Factors involving physical activity, static behavior, and sleep are associated with adolescent Internet addiction, and each item is also independently related to adolescent Internet addiction, especially physical activity. In the future, we must pay close attention to the Internet addiction of adolescents and take relevant measures, while increasing the physical activity of adolescents. It is necessary to reduce static behavior and ensure sufficient sleep, to reduce the Internet addiction of Chinese adolescents.

## Figures and Tables

**Table 1 ijerph-19-16873-t001:** The basic characteristics of the sample studied in this section.

Content	Options	Sample Size	Percentage
Sex	Boys	1156	56.5
	Girls	889	43.5
Grade	Fourth grade of elementary school	921	45.0
	Fifth-grade elementary school	1124	55.0
Only one child	Yes	511	25.0
	No	1534	75.0
Father’s level of education	Junior high school and below (including junior high school)	427	20.9
	High school/technical secondary school/vocational high school/technical school	491	24.0
	University/College/Higher Education	843	41.2
	Master’s or Ph. D	86	4.2
	Unknow	198	9.7
Mother’s level of education	Junior high school and below (including junior high school)	482	23.6
	High school/technical secondary school/vocational high school/technical school	511	25.0
	University/College/Higher Education	800	39.1
	Master’s or Ph. D	61	3.0
	Unknow	191	9.3
Whether or not to live with parents	Yes	1936	94.7
	No	109	5.3

**Table 2 ijerph-19-16873-t002:** Basic information of children reached the recommended amount of 24-h activity behavior guidelines.

Content	Options	n	%
Sleep guideline	Met	1473	72.0
	Not meet	572	28.0
Screen Time guideline	Met	780	38.1
	Not met	1265	61.9
MVPA guideline	Met	1807	88.4
	Not met	238	11.6
Number of guidelines	0	517	25.3
	1	1047	51.2
	2	415	20.3
	3	66	3.2
Guide combination	None	517	25.3
	Sleep guideline only	196	9.6
	Screen time guideline only	802	39.2
	Physical activity guideline only	49	2.4
	Sleep + screen time guideline	292	14.3
	Sleep + physical activity guideline	18	0.9
	Screen time + physical activity guideline	105	5.1
	Screen time + sleep + physical activity guideline	66	3.2

**Table 3 ijerph-19-16873-t003:** 24-h activity behavior guidelines (quantity) and logistic regression results for internet addiction.

Number of Reached Recommended Amount (s)	Odds Ratio	95% CI
None of them reached	8.46	3.06–23.36
One	4.50	1.64–12.39
Two	0.92	0.85–0.99
sleep, screen time, and MVPA	Reference	

**Table 4 ijerph-19-16873-t004:** 24-h activity behavior guidelines (different combinations) and logistic regression results for internet addiction.

Type of Reached Recommended Amount (s)	Odds Ratio	95% CI
None of three guidelines	8.44	3.06–23.31
Screen time and sleep	2.35	0.89–7.23
Sleep and MVPA	5.62	1.57–20.16
Screen time, and MVPA	4.33	1.45–12.87
Sleep only	4.05	1.42–11.56
Screen time only	4.43	1.60–12.23
MVPA only	7.29	2.37–22.40
sleep, screen time, and MVPA	Reference	

## Data Availability

The original contributions presented in this study are included in the article, further inquiries can be directed to the corresponding author.

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
