# Peer review of "Associations between 24-h Movement Behavior and Internet Addiction in Adolescents: A Cross-Sectional Study"

_ijerph, 2022, doi:10.3390/ijerph192416873_

Round 1

Reviewer 1 Report

In this paper, the authors aim to find out whether there is a relationship between 24-hour movement behaviour and internet addiction among Chinese adolescents. The study shows that adolescents have a sedentary lifestyle and less physical activity than desirable, which is related to internet addiction. The paper is particularly interesting because the 24-hour movement behaviour includes different variables such as sleep, sedentary behaviour and physical activity. In addition, they replicate similar studies carried out with some of the variables in other parts of Asia, obtaining similar results. The sample size of the students surveyed is also noteworthy.

Aspects to improve.

In my opinion, there are some issues that it would be advisable to correct before publishing the article: 

- Lines 32-36: some bibliographical reference to support these statements is missing.

- Lines 36-37: the definition of internet addiction is too brief and incomplete. As one of the main terms in the article, I think it is necessary to specify in more detail what is considered internet addiction.

- The meaning of the acronyms PA (line 106), MVPA (line 109), ST (line 115), SLP (line 126) should be indicated.

- Lines 132 to 139: What scale has been used to assess internet addiction? The authors cite reference no. 23, but, after consulting it, I have not found this scale.

- Table 2: repeated text "number of guides" and "guide combination".

Author Response

Responses to Reviewer 1 Comments

Dear reviewers,

Thank you for your time and valuable comments. We have provided a point-by-point response to each of your comments and suggestions and have made the appropriate changes to the manuscript. We believe the paper has improved significantly because of the review process.

Response to Reviewer comments

In this paper, the authors aim to find out whether there is a relationship between 24-hour movement behaviour and internet addiction among Chinese adolescents. The study shows that adolescents have a sedentary lifestyle and less physical activity than desirable, which is related to internet addiction. The paper is particularly interesting because the 24-hour movement behaviour includes different variables such as sleep, sedentary behaviour and physical activity. In addition, they replicate similar studies carried out with some of the variables in other parts of Asia, obtaining similar results. The sample size of the students surveyed is also noteworthy.

Aspects to improve.

In my opinion, there are some issues that it would be advisable to correct before publishing the article:

- Lines 32-36: some bibliographical reference to support these statements is missing.

Response: Thank you for your suggestion, please see line 40-41.

- Lines 36-37: the definition of internet addiction is too brief and incomplete. As one of the main terms in the article, I think it is necessary to specify in more detail what is considered internet addiction.

Response: Thank you for your suggestion, please see line 42-50.

- The meaning of the acronyms PA (line 106), MVPA (line 109), ST (line 115), SLP (line 126) should be indicated.

Response: Thank you for your suggestion, please see line 122, 126, 130 and 142.

- Lines 132 to 139: What scale has been used to assess internet addiction? The authors cite reference no. 23, but, after consulting it, I have not found this scale.

Response: Thank you for your suggestion, please see line 151.

- Table 2: repeated text "number of guides" and "guide combination".

Response: Thank you for your suggestion, please see the table 2.

Reviewer 2 Report

Over the past decade, an increasing number of studies have been highlighted the health benefits of high physical activity, low sedentary behaviour, and sufficient sleep. It has been shown that specific combinations of these behaviours are associated with desirable health indicators. A technique new to the health field for analysing 24-h movement behaviours data—compositional data analysis—has recently been introduced. And the authors follow this one. The composition of movement behaviours within a 24-h period may have important implications for health at all ages. There are many studies of associations between isolated forms of 24-h behaviour (sleep, sedentary behaviour, and physical activity) and internet addiction, but compositional analysis is rarely used. So, the research done is topical and presenting valuable empirical data for improving health promotion. The research is based upon the reliable and adequate assessment instruments, particularly the questionnaire which considers the actual devices for assessing screen time in adolescents. The empirical data are the most useful piece of the text. I have drawn my attention on the fact that adolescents who reach the only recommend amount of medium- and high-intensity physical activity per day or reach the recommend amount of both sleep and physical activity demonstrate relatively high probability of Internet addiction (7.29 and 5.62 correspondingly) in comparison with those who met none of the three guidelines for daily individual activity behaviours (8.44). And those who meet the recommend amount of sleep and screen time have the lowest probability of Internet addiction (2.35). The Odds ratio obtained allows to propose the hierarchy of recommended priorities for health promotion campaigns focusing on protective and harmful compositions of movement behaviours. And such recommendations may be expressed in Conclusion. I did not find such kind of data analysis and relevant conclusions. In Results section, there is not any analysis of the data presented in the Figures 1 and 2, the text merely duplicates the graphics in words and have no elucidations or explanations. The Introduction is theoretically insufficient and lacks any explanatory model relevant to the ideas written in Discussion: what are the reasons to relate empirical data obtained to family and school backgrounds, insufficient discipline provided the children by their parents? What are exact mechanisms through which 24-h movement behaviours may be related to Internet addiction? It needs to be modelled or revealed in Introductory section.

The whole text is slipshod. The lines 32-35 are repetitious: significant use and significant increase (32), the 47-th most recent statistics (33) and the 47-th latest statistics (35). The measurement of socio-economic status includes only parents’ education level, but in Results the authors write about index of affluence (money, property, wealth – where are these indicators of affluence in the description of the sample?!). In line 72 the numbers 50.4 % and 21.7 % do not correspond with the numbers in Table 2: 51.2 % and 20.3 %. In line 188, I suppose that it comes to “one item recommended amount”, not “a recommended amount”. In general, I think Introduction is better to improve with description of theoretical model, Results may be improved with analysis and comments of the empirical data obtained. And in Discussion make the salient explanation of why adolescents who reached the standard of medium- and high-intensity activity turned to be more prone to Internet addiction if past studies have shown a negative correlation between physical activity levels and internet addiction (lines 44-45 of the article).

Author Response

Dear reviewers,

Thank you for your time and valuable comments. We have provided a point-by-point response to each of your comments and suggestions and have made the appropriate changes to the manuscript. We believe the paper has improved significantly because of the review process.

Response to Reviewer comments

Reviewer 2

Over the past decade, an increasing number of studies have been highlighted the health benefits of high physical activity, low sedentary behaviour, and sufficient sleep. It has been shown that specific combinations of these behaviours are associated with desirable health indicators. A technique new to the health field for analysing 24-h movement behaviours data—compositional data analysis—has recently been introduced. And the authors follow this one. The composition of movement behaviours within a 24-h period may have important implications for health at all ages. There are many studies of associations between isolated forms of 24-h behaviour (sleep, sedentary behaviour, and physical activity) and internet addiction, but compositional analysis is rarely used. So, the research done is topical and presenting valuable empirical data for improving health promotion. The research is based upon the reliable and adequate assessment instruments, particularly the questionnaire which considers the actual devices for assessing screen time in adolescents. The empirical data are the most useful piece of the text. I have drawn my attention on the fact that adolescents who reach the only recommend amount of medium- and high-intensity physical activity per day or reach the recommend amount of both sleep and physical activity demonstrate relatively high probability of Internet addiction (7.29 and 5.62 correspondingly) in comparison with those who met none of the three guidelines for daily individual activity behaviours (8.44). And those who meet the recommend amount of sleep and screen time have the lowest probability of Internet addiction (2.35). The Odds ratio obtained allows to propose the hierarchy of recommended priorities for health promotion campaigns focusing on protective and harmful compositions of movement behaviours. And such recommendations may be expressed in Conclusion. I did not find such kind of data analysis and relevant conclusions. In Results section, there is not any analysis of the data presented in the Figures 1 and 2, the text merely duplicates the graphics in words and have no elucidations or explanations. The Introduction is theoretically insufficient and lacks any explanatory model relevant to the ideas written in Discussion: what are the reasons to relate empirical data obtained to family and school backgrounds, insufficient discipline provided the children by their parents? What are exact mechanisms through which 24-h movement behaviours may be related to Internet addiction? It needs to be modelled or revealed in Introductory section.

The whole text is slipshod. The lines 32-35 are repetitious: significant use and significant increase (32), the 47-th most recent statistics (33) and the 47-th latest statistics (35). The measurement of socio-economic status includes only parents’ education level, but in Results the authors write about index of affluence (money, property, wealth – where are these indicators of affluence in the description of the sample?!). In line 72 the numbers 50.4 % and 21.7 % do not correspond with the numbers in Table 2: 51.2 % and 20.3 %. In line 188, I suppose that it comes to “one item recommended amount”, not “a recommended amount”.

Response: Thank you for your suggestion, please see line 37-40, 119, 190 and 190-191.

In general, I think Introduction is better to improve with description of theoretical model, Results may be improved with analysis and comments of the empirical data obtained.

Response: Thank you for your suggestion, please see line 33-50, 243-252.

And in Discussion make the salient explanation of why adolescents who reached the standard of medium- and high-intensity activity turned to be more prone to Internet addiction if past studies have shown a negative correlation between physical activity levels and internet addiction (lines 44-45 of the article).

Response: Thank you for your suggestion, however, in this manuscript, we use a better combination as a reference. For example, three behaviors, sufficient physical activity, less screen time, and adequate sleep, this is the optimal combination. Then any one of the following optimal combinations, such as more physical activity, better sleep, or less static behavior, may lead to Internet addiction. For example, less static behavior, or adequate sleep, this alone In other cases, there may be more cases of Internet addiction.
